# Blueberry–Mulberry Extract Alleviates Cognitive Impairment, Regulates Gut Metabolites, and Inhibits Inflammation in Aged Mice

**DOI:** 10.3390/foods12040860

**Published:** 2023-02-17

**Authors:** Hui Li, Changhao Xiao, Feng Wang, Xuqi Guo, Zhongkai Zhou, Yugang Jiang

**Affiliations:** 1Institute of Environmental and Operational Medicine, Tianjin 300050, China; 2Key Laboratory of Food Nutrition and Safety, Ministry of Education, Tianjin University of Science and Technology, Tianjin 300457, China

**Keywords:** polyphenol-rich extract, cognition, gut microbiota, gut metabolites, naturally aging mice

## Abstract

Cognitive impairment is associated with aging; however, the underlying mechanism remains unclear. Our previous study found that polyphenol-rich blueberry–mulberry extract (BME) had an antioxidant capability and effectively alleviated cognitive impairment in a mouse model of Alzheimer’s disease. Thus, we hypothesized that BME would improve cognitive performance in naturally aging mice and assessed its effects on related signaling pathways. Eighteen-month-old C57BL/6J mice were gavaged with 300 mg/kg/d of BME for 6 weeks. Behavioral phenotypes, cytokine levels, tight junction protein levels, and the histopathology of the brain were assessed, and 16S ribosomal RNA sequencing and targeted metabolome analyses were used for gut microbiota and metabolite measurements. Our results showed that the cognitive performance of aged mice in the Morris water maze test was improved after BME treatment, neuronal loss was reduced, IL-6 and TNF-α levels in the brain and intestine were decreased, and the levels of intestinal tight junction proteins (ZO-1 and occludin) were increased. Further, 16S sequencing showed that BME significantly increased the relative abundance of *Lactobacillus*, *Streptococcus,* and *Lactococcus* and decreased the relative abundance of *Blautia*, *Lachnoclostridium,* and *Roseburia* in the gut. A targeted metabolomic analysis showed that BME significantly increased the levels of 21 metabolites, including α-linolenic acid, vanillic acid, and N-acetylserotonin. In conclusion, BME alters the gut microbiota and regulates gut metabolites in aged mice, which may contribute to the alleviation of cognitive impairment and to inflammation inhibition in both the brain and the gut. Our results provide a basis for future research on natural antioxidant intervention as a treatment strategy for aging-related cognitive impairment.

## 1. Introduction

Cognitive impairment in the elderly is closely related to neurodegenerative diseases, especially Alzheimer’s disease (AD), which seriously reduces their quality of life and places a heavy burden on their families and society [1,2]. The pathology of cognitive impairment in the elderly is associated with chronic inflammation, which impairs neurogenesis and neural stem cell survival and differentiation and promotes the development of neurodegenerative diseases. Studies have shown that cytokines, including IL-1β, TNF-α, and IL-6, in the peripheral blood of participants with mild cognitive impairment were increased compared with those of controls [3,4]. Chronic neuroinflammation is associated with the breakdown of the blood–brain barrier, which permits serum lipopolysaccharides (LPS) to pass into the brain and leads to the activation of microglia [5,6], resulting in increased reactive oxygen species (ROS) production and cytokine expression in the hippocampus. These factors prohibit neuronal repair, resulting in synaptic impairment, oxidative damage, and mitochondrial dysfunction, leading to neurodegeneration [7].

Currently, increasing emphasis has been placed on the role of the gut microbiota in modulating brain function and behavior. The microbiota–gut–brain axis has been proposed as a bidirectional communication pathway enabling gut microbes to communicate with the brain [8]. With increasing age, the composition of the gut microbiota changes significantly, and the extent of the gut microbiota diversity correlates with the health of the elderly [9]. A study found that germ-free mice that received gut microbiota from aged mice demonstrated depressive-like behavior, impaired short-term memory, and impaired spatial memory over the 3 months following the initial fecal transplant gavages [10]. Importantly, the microbiota in the elderly were shown to be strongly influenced by diet [11], and dietary intervention is considered the main strategy for shaping the gut microbiota in this population [12].

Polyphenols are the main bioactive components of blueberries and mulberries, exerting antioxidant and anti-inflammatory effects. Several studies found that polyphenols effectively alleviated disorders of free radicals, enhanced immunity, reduced cytokine levels caused by aging, and consequently increased neural activity and improved working memory performance [13]. In our previous study, we demonstrated that blueberry extract improved the performance of APP/PS1 mice in the Morris water maze (MWM) test and enhanced long-term potentiation (LTP) [14]. We also found that mulberry extract alleviated Aβ_25-35_-induced injury in PC12 cells [15]. Furthermore, both blueberry and mulberry extracts changed a gut microbiota that was disturbed by a high-fat diet in mice and reduced the levels of intestinal cytokines, such as TNF-α and IL-1β [16,17]. However, the mechanisms by which polyphenol-rich extracts alleviate cognitive decline in naturally aging mice remains unclear.

In this study, we hypothesized that blueberry–mulberry extracts (BME) could alleviate cognitive decline in naturally aging mice and investigated the behavioral phenotypes, histopathology of the brain, gut function, microbiota composition, and metabolite levels in aged mice treated with BME. This study aimed to serve as a basis to provide a new strategy for nutritional intervention against aging-related cognitive decline in the elderly.

## 2. Methods and Materials

### 2.1. Materials

Blueberries (*Vaccinium uliginosum* L.) were purchased from Daxinganling Lingongberry Organic Foodstuffs Co. Ltd., Harbin, Heilongjiang Province, China. Fresh blueberries were crushed and extracted four times with fresh ethanol (75%) at room temperature for 96 h. The four ethanolic phases were recovered, pooled, dried at 40 °C in a rotary evaporator, and lyophilized [14]. Mulberries (*Morus nigra* L.) were purchased from Xinjiang Bencaotang Traditional Chinese Herbal Decoction Pieces Co. Ltd., Turfan, Xinjiang Uygur Autonomous Region, China. Blended fresh mulberries were extracted five times with 60% alcohol, dehydrated in a rotary evaporator, and lyophilized [15]. A BME stock solution was prepared by dissolving blueberry extract and mulberry extract (1:1) in water for gavage.

### 2.2. Animals and Treatments 

Eighteen-month-old male C57BL/6J mice and six-month-old male C57BL/6J mice were purchased from Beijing SPF Bioscience Co. Ltd. (Beijing, China). Mice were maintained in the laboratory animal center of the Tianjin Institute of Environmental and Operational Medicine under standard housing conditions (ambient temperature of 23 ± 2 °C and 50–60% humidity on a 12 h light/dark cycle). All animals were allowed to adapt to the laboratory environment for 4 days before the experiment and were provided with an AIN-93 diet (Table 1) and water ad libitum. The 18-month-old mice were randomly divided into two groups (*n* = 8 in each group): an aged control group (AC) that was gavaged with 0.2 mL of sterile saline and a BME treated group (BME) that was gavaged with 300 mg/kg/d of BME. The 6-month-old mice were chosen as the young control group (YC) and were gavaged with 0.2 mL of sterile saline. Subsequently, the mice were sacrificed and brain samples and cecal contents were collected at the end of 6 weeks, after the behavioral evaluation. All animal experimental protocols were approved by the Institutional Animal Care and Use Committee of the Tianjin Institute of Environmental and Operational Medicine (AMMS-04-2020-010).

### 2.3. Morris Water Maze Test

The MWM test has been widely used to evaluate the spatial memory abilities of mice, as previously described [14]. The MWM was composed of an open circular pool (120 cm in diameter) containing warm water and equipped with animal behavior video analysis software (SuperMaze software, Shanghai, China). The pool was divided into four imaginary quadrants, and a platform (12 cm in diameter) was placed at 1 cm under the water surface in one of the quadrants (the target quadrant). The MWM test was composed of a training phase and a probe test. In the training phase, the mice were trained to seek the hidden platform within 60 s over a period of 4 days. On the fifth day, the platform was removed, and each mouse was allowed to swim freely for 60 s in the probe test. The number of times the mouse passed across the platform and the time spent in the target quadrant were recorded.

### 2.4. Open Field Test

The open field test was used to evaluate the exploratory behavior of mice, as previously described [18]. Briefly, mice were allowed to freely explore a test box (50 cm × 50 cm × 50 cm) for 3 min. The total distance traveled, the time spent not moving, and the total crossing scores (the total number of zones that were crossed by the subjects) were recorded using animal behavior video analysis software. After each mouse was tested, the apparatus was cleaned with a 75% ethanol solution.

### 2.5. Enzyme-Linked Immunosorbent Assay (ELISA)

Brain tissues were ground in saline in a chilled tissue homogenizer, and the supernatant was collected after centrifugation at 13,000× *g* for 10 min. The expression levels of IL-2 (EK202/2, Multi sciences (Lianke) Biotech, Hangzhou, China), IL-6 (EK206/3, Multi sciences (Lianke) Biotech, Hangzhou, China), and TNF-α (EK282/3, Multi sciences (Lianke) Biotech, Hangzhou, China) were measured using ELISA kits according to the manufacturer’s instructions.

### 2.6. Hematoxylin and Eosin (HE) Staining

HE staining was performed on the brain tissue using a previously described procedure [19]. The stained sections were observed under an Olympus BX61 light microscope (Olympus, Tokyo, Japan). Images were captured using an Olympus DP70 digital camera (Olympus, Tokyo, Japan). To assess necrotic neurons in the CA3 region, shrunken neurons with pyknotic nuclei were considered to be necrotic neurons. Image Pro Plus (version 6.0, Rockville, MD, USA) was used to count the number of necrotic neurons per field under a light microscope. Data were expressed as the number of necrotic neurons/field, as previously reported [20].

### 2.7. Immunohistochemistry Staining

Brain and colon tissues were paraffin-embedded, processed into 5 μm thick slices, and dewaxed in an organic solvent concentration gradient. Tissue sections were placed in a microwave oven for antigen retrieval, immersed in a 3% hydrogen peroxide solution at room temperature, and protected from light for 25 min to block endogenous peroxidase. After the slices were dried, primary antibodies against IL-6 (bs-0782R, Bioss, Beijing, China), IL-10 (bs-0698R, Bioss, Beijing, China), TNF-α (bs-10802R, Bioss, Beijing, China), ZO-1 (ab96587, Abcam, Cambridge, UK), and occludin (ab216327, Abcam, Cambridge, UK) were diluted in 3% normal goat serum and applied to the tissue. The sections were placed in a wet box and incubated overnight at 4 °C. After washing, an HRP-conjugated secondary antibody (ZSGB Biotech, Beijing, China) was applied to the tissue. The nuclei were stained with hematoxylin, and the tissue sections were sealed with neutral gum. Images were captured using an Olympus BX61 light microscope. The digital images were analyzed using Image Pro Plus (version 6.0), as previously described [21,22]. The gray unit was switched to the optical density (OD) unit in the software, and the integrated OD analysis was performed with immunohistochemically stained regions.

### 2.8. Sequencing of the 16S Ribosomal RNA Gene in the Microbiota

The 16S ribosomal RNA gene (16S rRNA) sequencing was performed according to our previous study [23]. Briefly, DNA was extracted from cecal content samples using the cetyltrimethylammonium bromide/sodium dodecyl sulfate method. The 16S rRNA V4 regions were amplified using specific primers (515F and 806R). All PCR reactions were performed using Phusion^®^ High-Fidelity PCR Master Mix (New England Biolabs, Ipswich, MA, USA). The 16S rRNA gene was analyzed to evaluate the bacterial diversity using IonS5^TM^XL (ThermoFisher, Waltham, MA, USA). Sequences were analyzed using the Cutadapt software (Version 1.9.1, http://cutadapt.readthedocs.io/en/stable/ (accessed on 20 April 2020) developed by Martin et al. [24]). The Shannon index was calculated as the alpha diversity, the unweighted UniFrac was used to calculate the beta diversity, and Bray–Curtis metrics were used for a principal coordinate analysis using R software (Version 2.15.3). The linear discriminant analysis effect size (LEfSe) [25] conducted on the website (Version 1.0, https://huttenhower.sph.harvard.edu/galaxy (accessed on 2 April 2020)) was adopted to search for the biomarkers of different groups. Based on the LEfSe analysis, bacteria with *p* values < 0.05 in the Kruskal–Wallis sum test and linear discriminant analysis (LDA) scores of 4.0 were plotted.

### 2.9. Targeted Metabolomics Profiling in Microbiota

Targeted metabolomic profiling of cecum content samples was performed by Metabo-Profile (Shanghai, China). The samples were prepared as described previously [26,27]. Briefly, 10 mg of lyophilized cecum content samples were homogenized with 25 μL of water and extracted with 185 μL of cold acetonitrile–methanol (8/2, *v*/*v*). After centrifugation, 30 μL of the supernatant was derivatized with 20 μL of freshly prepared derivative reagents on a Biomek 4000 workstation (Biomek 4000, Beckman Coulter, Inc., Brea, CA, USA), followed by mixing with internal standards. All 310 standard substances were obtained from Sigma-Aldrich (St. Louis, MO, USA), Steraloids Inc. (Newport, RI, USA) and TRC Chemicals (Toronto, ON, Canada).

The samples and serial dilutions of derivatized stock standards were randomly analyzed and quantitated using ultra-performance liquid chromatography coupled with a tandem mass spectrometry (UPLC-MS/MS) system (ACQUITY UPLC-Xevo TQ-S, Waters Corp., Milford, CT, USA). Test mixtures, internal standards, and pooled biological samples are routinely used in metabolomics platforms. The derivatized pooled quality control samples were injected every 14 test samples. QuanMET software (v2.0, Metabo-Profile, Shanghai, China) was used to process raw data for the peak integration, calibration, and quantification of each metabolite. R software was used to analyze the differential metabolites screened by the *t*-test (*p* < 0.05), and the results are shown in heat and volcano plots. An orthogonal partial least squares discriminant analysis and a variable importance in projection (VIP) analysis were performed, and differential metabolites with VIP > 1 were screened for an enrichment analysis with pathway-associated metabolite sets.

### 2.10. Statistical Analyses

Data are presented as means ± standard errors of means (SEMs). After checking the normality of the data, differences between the means were analyzed using SPSS 20.0 (IBM, Armonk, NY, USA). An analysis of variance followed by a least significant difference test was used for multiple comparisons. Statistical significance was set at *p* value < 0.05.

## 3. Results 

### 3.1. BME Improved the Cognitive Performance of Aged Mice in the MWM Test

The MWM and open field tests were performed after 6 weeks of BME treatment to investigate the effect of BME on cognitive performance in aged mice. In the training phase (Figure 1A), compared with the YC group, the latency of mice was increased in the AC group and decreased (*p* < 0.05) in the BME group on days 3 and 4. In the probe test (Figure 1B), compared with the YC group, crossing the platform decreased (*p* < 0.05) in the AC group and increased after BME treatment. These results demonstrate that spatial memory in aged mice was impaired and BME alleviated this impairment.

The results of the open field test showed that the total distance traveled (Figure 1C) and the total crossing scores of mice in the AC group were decreased (Figure 1D, *p* < 0.05) whereas the time spent not moving was increased compared to that of the YC group (Figure 1E, *p* < 0.05). After the BME treatment, the total distance traveled, the time spent not moving, and the total crossing scores of aged mice showed no significant changes (*p* > 0.05). Therefore, the findings suggest that BME does not promote exploratory behavior.

### 3.2. BME Inhibited Brain Inflammation and Neuronal Loss in Aged Mice

To study the effect of BME on the brain tissue morphology of aged mice, HE staining was performed. Neuronal loss was observed in the pyramidal layer of the hippocampus of aged mice (Figure 2A,B), as necrotic neurons were stained “Modena” due to the presence of pyknotic nuclei. The assessment of pyknotic nuclei also indicated a significant loss of neurons in the AC group compared with that in the YC group (*p* < 0.05), while the BME treatment significantly decreased the number of necrotic neurons (*p* < 0.05).

We performed immunohistochemistry on the cerebral cortex and ELISA on whole brain lysates to investigate the effects of aging and BME treatment on inflammation. Aging increased the levels of the pro-inflammatory cytokines IL-6 (Figure 2C,D) and TNF-α (Figure 2G,H) in the cerebral cortex (*p* < 0.05) and whole brain lysates (Figure 3B,C, *p* < 0.05). The level of the anti-inflammatory cytokine IL-10 was decreased in the cerebral cortex of mice in the AC group (Figure 2E,F, *p* < 0.05). Meanwhile, the BME treatment exhibited anti-inflammatory effects, which effectively reversed the increased levels of pro-inflammatory cytokines, including IL-6, IL-2, and TNF-α, and decreased the levels of the anti-inflammatory cytokine IL-10 (*p* < 0.05).

### 3.3. BME Reduced Intestinal Inflammation and Improved Intestinal Barrier Function in Aged Mice

The levels of pro-inflammatory and anti-inflammatory cytokines in the intestine were detected using immunohistochemistry. The results showed that IL-6 (Figure 4A,B), IL-10 (Figure 4C,D), and TNF-α (Figure 4E,F) were mainly expressed in the intestinal and glandular epithelial cells. Aging elevated the levels of pro-inflammatory cytokines, including IL-6 and TNF-α (*p* < 0.05), and decreased the level of the anti-inflammatory cytokine IL-10 (*p* < 0.05). The BME group also had significantly decreased IL-6 and TNF-α levels and increased IL-10 levels compared to the AC group (*p* < 0.05). 

Cytokines modulate intestinal permeability by regulating tight junction protein expression and trafficking [28,29]. To evaluate the intestinal permeability, we detected the levels of tight junction proteins using immunohistochemistry. The results showed that the levels of ZO-1 and occludin were inhibited by aging (Figure 5, *p* < 0.05), suggesting an abnormality in intestinal permeability. The BME treatment elevated the expression levels of ZO-1 and occludin in aged mice (*p* < 0.05), suggesting that BME reduced intestinal inflammation and protected intestinal permeability in these mice.

### 3.4. BME Altered the Gut Microbiota in Aged Mice

The effects of the BME treatment on gut microbiota were determined using a 16S rRNA sequencing-based analysis. An alpha diversity analysis showed that there was no significant difference between the YC group and AC groups for the Shannon index (Figure 6A, *p* < 0.05) and that the BME group presented significantly decreased microbial diversity compared to the AC group (*p* < 0.05). However, aging decreased the beta diversity, and the BME treatment reversed this effect (Figure 6B, *p* < 0.05). Furthermore, the taxonomic composition was clearly distinguished among the three groups (Figure 6C). These results suggest that both aging and the BME treatment altered the gut microbiota. 

The taxonomic profiling showed that the microbiota composition was altered in the AC and BME groups at the phylum and genus levels (Figure 6D,E). At the genus level, the relative abundances of *Allobaculum, Blautia, Romboutsia, Turicibacter, Clostridiales,* and *Anaerotruncus* in the AC group (Figure 6F) were significantly increased (*p* < 0.05), whereas those of *Bacteroides, Alloprevotella,* and *Odoribacter* were significantly decreased compared to the YC group (*p* < 0.05). At the genus level, the BME treatment increased the relative abundances of *Lactobacillus, Streptococcus, Lactococcus, Corynebacteriaceae, Aerococcus, Enterococcus, Leuconostoc,* and *Weissella* (Figure 6G, *p* < 0.05), whereas those of *Blautia, Lachnoclostridium, Roseburia,* and *Anaerotruncus* were significantly decreased (*p* < 0.05). Moreover, to further characterize the changes in the gut microbiota, an LEfSe analysis was used to gain insight into the differences among the groups (Figure 6H,I). The results showed that there were three significantly different classes: Bacteroidia, which was the highest in the YC group; Bacilli, which was enriched in the BME group; and Clostridia, which was enriched in the AC group. According to the LEfSe analysis, these abundant taxa could be considered potential biomarkers (LDA score 4.0, *p* < 0.05). Potential biomarkers at the genus level included *Faecalibaculum* in the YC group, *Lactobacillus* in the BME group, and *Blautia* in the AC group.

### 3.5. BME Alters Metabolites of Gut Microbiota in Aged Mice

The effects of the BME treatment on the metabolites of gut microbiota were determined by targeted metabolomic profiling, and a total of 150 metabolites were identified. The results showed that, compared with the AC group, 21 intestinal metabolites were significantly increased and two metabolites were significantly decreased in the BME treatment group (Figure 7A,B, *p* < 0.05), including nine fatty acids (alpha-linolenic acid, eicosapentaenoic acid (EPA), linolenic acid, myristoleic acid, octanoic acid, oleic acid, palmitoleic acid, and pentadecanoic acid), two amino acids (cystine and methylcysteine), two benzoic acids (4-aminohippuric acid and vanillic acid), two carnitines (carnitine and propionylcarnitine), and two indoles (indole-3-propionic acid and N-acetylserotonin (NAS)), glucose, oxoadipic acid, homovanillic acid, nicotinic acid, and caproic acid.

The enrichment analysis of metabolites (Figure 7C) showed that the differential metabolites were mainly enriched in the terms “beta oxidation of long-chain fatty acids,” “metabolism of Alpha linolenic acid and linoleic acid,” and “oxidation of short-chain saturated fatty acids of mitochondria.”

Moreover, we investigated the correlations between the altered operational taxonomic units (OTUs) and metabolites using a Spearman’s correlation analysis. The results showed (Figure 7D) that 15 metabolites were correlated with the abundances of the three microbiota taxa (*p* < 0.05, r > 0.3). Specifically, *Blautia* was positively correlated with methylcysteine but negatively correlated with vanillic acid, 4-aminohippuric acid, oxoadipic acid, NAS, nicotinic acid, and linoleic acid; *Lactobacillus* was negatively correlated with methylcysteine and positively correlated with vanillic acid, 4-aminohippuric acid, oxoadipic acid, NAS, nicotinic acid, gamma-linolenic acid, linoleic acid, and pentadecanoic acid; and *Streptococcus* was negatively correlated with methylcysteine and positively correlated with nicotinic acid, linoleic acid, gamma-linolenic acid, EPA, caproic acid, octanoic acid, alpha-linolenic acid, and homovanillic acid. These data suggest that the gut microbiota composition is closely related to the metabolites that are present, which may contribute to the alteration of gut function. 

## 4. Discussion

In this study, we found that BME changed the gut microbial community and metabolites, restored intestinal epithelial barrier function, suppressed inflammation, and improved the cognitive performance of aging mice, which is consistent with our hypothesis. Thus, the BME treatment could potentially be a strategy for nutritional intervention against cognitive impairment in the elderly.

Low-grade inflammation, a main characteristic of the brain aging process, can be reduced by activated macrophages and monocytes. The accumulation of ROS leads to lymphocyte apoptosis, gradually reducing immune function and accelerating the aging process [30]. In this study, it was found that the spatial memory ability of mice was decreased in the elderly group compared to that in the young group, and the number of necrotic neurons in the hippocampus as well as the levels of related inflammatory factors in the brain, such as IL-6 and TNF-α, were significantly increased, suggesting that the decline in cognitive function in the elderly mice was closely related to the inflammation. The BME treatment alleviated the cognitive impairment and reduced the levels of related cytokines in the brains of aging mice. Indeed, blueberries and mulberries are believed to have the ability to provide cellular antioxidant protection, scavenge free radicals, inhibit inflammatory gene expression, and consequently protect against oxidant-induced cell damage and cytotoxicity [31,32,33]. Numerous studies have indicated that anthocyanins, the main contents of BME, may play crucial roles in the prevention and treatment of different pathological conditions, which have encouraged their consumption around the world [34]. Anthocyanins exhibit a significant neuroprotective role, mainly due to their well-recognized antioxidant and anti-inflammatory properties. However, which anthocyanin monomer is beneficial for cognitive improvement needs further research.

Intestinal epithelial barrier dysfunction is closely associated with inflammation. Tight junctions, consisting of transmembrane proteins (e.g., occludin) and cytoplasmic membrane proteins (e.g., ZO-1 and ZO-2), play a major role in the functional maintenance of the intestinal epithelial barrier. During the aging process, the decreased expression of tight junction proteins disturbs the permeability of the intestinal epithelial barrier, leading to toxic substances crossing the intestinal mucosa and causing local or even systemic inflammatory responses [28]. In this study, we found that the intestinal epithelial barrier was weakened and that cytokines were produced during the aging process. The BME treatment promoted the expression of intestinal barrier proteins and reversed intestinal inflammation. Cheng et al. confirmed that fermented blueberry pomace improved the intestinal morphology and barrier function in mice fed a high-fat diet, including increased ZO-1 and occludin levels in the intestine and decreased TNF-α levels and increased IL-10 levels in the serum [35]. Furthermore, malvidin-3-glucoside, an anthocyanin found in blueberries, was shown to alleviate gut dysbiosis and reverse elevated levels of several key inflammatory mediators, including sphingolipid metabolites, in dextran sulfate sodium colitis mice [36]. These results suggest that BME restored the intestinal epithelial barrier.

As an important communication tool between the host immune system and the commensal microbiota, gut bacterial metabolites affect brain inflammation and function [37,38]. In our study, we found that the BME treatment significantly increased the levels of vanillic acid, NAS, and polyunsaturated fatty acids (PUFAs), including α-linolenic acid, γ-linolenic acid, EPA, and linoleic acid, in the intestine. PUFAs are major phospholipid constituents of the membranes of cells involved in the immune response and in the maintenance of brain health. Recent studies have focused on the molecular signaling regulation of microglia by PUFAs, especially in the context of neuroinflammation and behavior [39]. Chen et al. showed that ω-3 PUFA supplementation promoted a shift from the M1 microglial phenotype to the M2 microglial phenotype and inhibited microglial activation, thus reducing traumatic brain injury induced inflammatory factors [40]. In addition, vanillic acid, a kind of natural polyphenolic substance with neuroprotective potential, is a BME metabolite that was found in the mouse intestines. Khoshnam et al. found that vanillic acid improved the performance of rats with ischemic stroke in the MWM test and reduced the levels of IL-6, TNF-α, and TUNEL-positive cells [41]. Amin et al. found that mice treated with vanillic acid had decreased Aβ_1-42_-induced neuronal apoptosis and neuroinflammation as well as improved synaptic function and alleviated cognitive deficits. They also demonstrated that vanillic acid was non-toxic to HT22 cells and increased cell viability after Aβ_1-42_ exposure [42]. Moreover, NAS, an intermediate product of melatonin biosynthesis from serotonin, exerts anti-inflammatory, antidepressant-like, and cognition-enhancing effects. Aging is associated with decreased NAS production, largely resulting from the downregulation of beta 1-adrenoreceptors that activate serotonin N-acetyltransferase, the enzyme catalyzing the formation of NAS from serotonin [43]. Bachurin found that an NAS treatment for 3 weeks effectively reversed the poor performance of mice induced by the cholinergic neurotoxin ethylcholine aziridinium in the active avoidance and MWM tests [44]. These results suggest that BME may influence cognition through an interaction with gut bacterial metabolites, especially PUFAs, vanillic acid, and NAS. 

The gut microbiota play an important role in metabolite production. We found that *Blautia* and *Allobaculum* were increased and *Bacteroides* were decreased in the aged mice. After the BME treatment, *Blautia* decreased, whereas *Lactobacillus, Lactococcus,* and *Streptococcus* increased. *Blautia* has been reported to be closely related to the metabolism of short-chain fatty acids, especially butyrate, which contributes to the integrity of the intestinal barrier and a reduction in intestinal inflammation [45]. However, in 6-month-old APP/PS1 mice, *Blautia* increased significantly, and AD pathological features, such as Aβ deposition, were also observed [46]. Vogt et al. found that *Blautia* was significantly increased in patients with AD and was positively correlated with the concentrations of p-tau and Aβ_42_ in the cerebrospinal fluid [47]. We also found that NAS was negatively correlated with *Blautia*; however, the relationship between NAS and *Blautia* has not been reported, highlighting a need for further study. *Allobaculum* is closely related to lipid metabolism, and its abundance was significantly increased in mice fed a high-fat diet. It was positively correlated with IL-6, IL-1β, and TNF-α levels [48]. These findings are consistent with our results and suggest that *Blautia* and *Allobaculum* may have negative effects on intestinal stability during aging. In the past, *Streptococcus* has always been considered an opportunistic pathogen; however, in recent years, studies have found that it can play an anti-inflammatory role. Indeed, Han et al. found that *Streptococcus thermophilus 19* significantly reduced the expression of TNF-α, IL-1β, and IL-6 in LPS-induced macrophages [49]. Albouery et al. found that dysregulated gut microbiota in old donor mice decreased the concentrations of PUFAs in the brains of germ-free mice [50]. Our results showed that PUFAs were positively correlated with *Streptococcus*, and the dysregulation of *Streptococcus* levels might be the reason for the decreased concentrations of PUFAs in aged mice, although this finding requires further confirmation. *Lactobacillus* has been widely used as a probiotic in food to maintain intestinal health [51]. We found that vanillic acid was positively correlated with *Lactobacillus,* and Zhang et al. confirmed that vanillic acid was the main component in *Lactobacillus plantarum* dy-1 fermented cereal [52]. Supplementation of *Lactobacillus plantarum* in APP/PS1 mice for 12 weeks effectively remodeled the gut microbiota, reduced hippocampal inflammation, ameliorated cognitive deterioration, and decreased the Aβ levels in the brain [53]. *Lactococcus* also contributes to the maintenance of body health; Zurita-Turk et al. found that *Lactococcus lactis* could inhibit the production of intestinal IL-6 and increase IL-10 in mice, thereby alleviating intestinal inflammation [54]. Therefore, the anti-inflammatory mechanism of BME could be indicated for regulating the gut microbiota and metabolites.

Anthocyanins are water-soluble flavonoid pigments that are widely found in plants, and their types and concentrations vary among fruit species and areas of origin, giving petals and fruits a variety of colors. Yang et al. reported that blueberry extract contains various anthocyanins, including cyanidin-3-O-glucoside (Cy3G), malvidin-3-O-glucose, pelargonidin-3-O-galactoside, and delphinidin-3-glucoside [55]. Mo et al. found that the main components of mulberry extract were Cy3G and cyanidin-3-O-rutinoside [56]. In our previous study, the content of Cy3G in the blueberry extract was about 25%, and that in the mulberry extract was 6.8% [15,57]. The biological activities of anthocyanins are closely related to their structures [58]. The mixing of two extracts with a variety of anthocyanins may be beneficial to their biological function.

In summary, this mouse study demonstrated that aging damaged cognition, altered the gut microbiota, deteriorated the intestinal barrier function, and promoted inflammation, whereas BME alleviated cognitive impairment, restored intestinal barrier function, and decreased inflammation. BME was also shown to alter the gut microbiota and metabolites, which may facilitate its positive effects. Next, we plan to investigate whether BME promotes cognition through the gut microbiota by transferring the microbiota into germ-free mice. 

## Figures and Tables

**Figure 1 foods-12-00860-f001:**
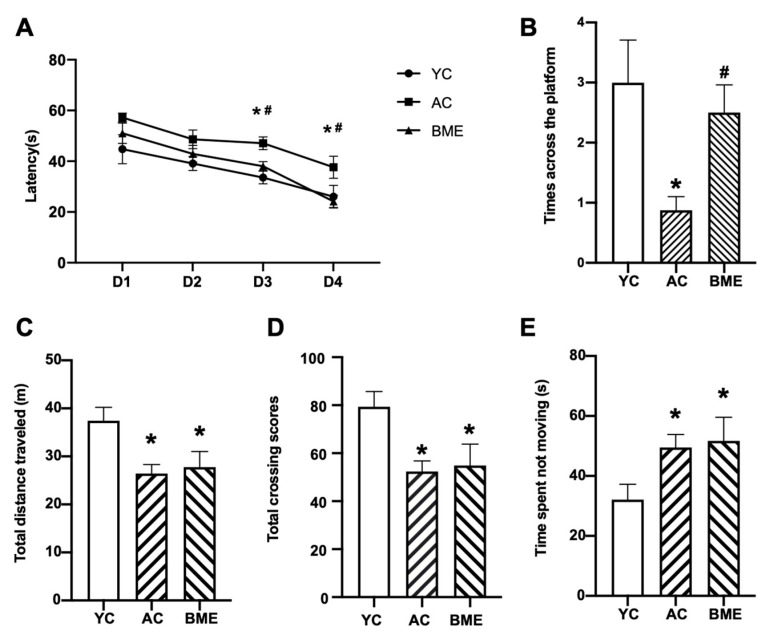
The effects of BME on the behaviors of aged mice. After BME and sterile saline treatment on 18-month-old (BME and AC group) and 6-month-old (YC group) mice for 6 weeks, the Morris water maze and open field tests were performed. (**A**) Escape latency of mice in the training phase of the Morris water maze test. (**B**) Number of times the mice crossed the platform after hiding the platform in the Morris water maze test. (**C**) The total distance that the mice traveled in the open field test. (**D**) Time spent not moving by mice in the open field test. (**E**) Total crossing scores (the total number of zones that were crossed) of mice in the open field test. All results are expressed as means ± SEMs (*n* = 8). * *p* < 0.05 versus YC group, # *p* < 0.05 versus AC group. AC: aged control, BME: blueberry–mulberry extracts, YC: young control.

**Figure 2 foods-12-00860-f002:**
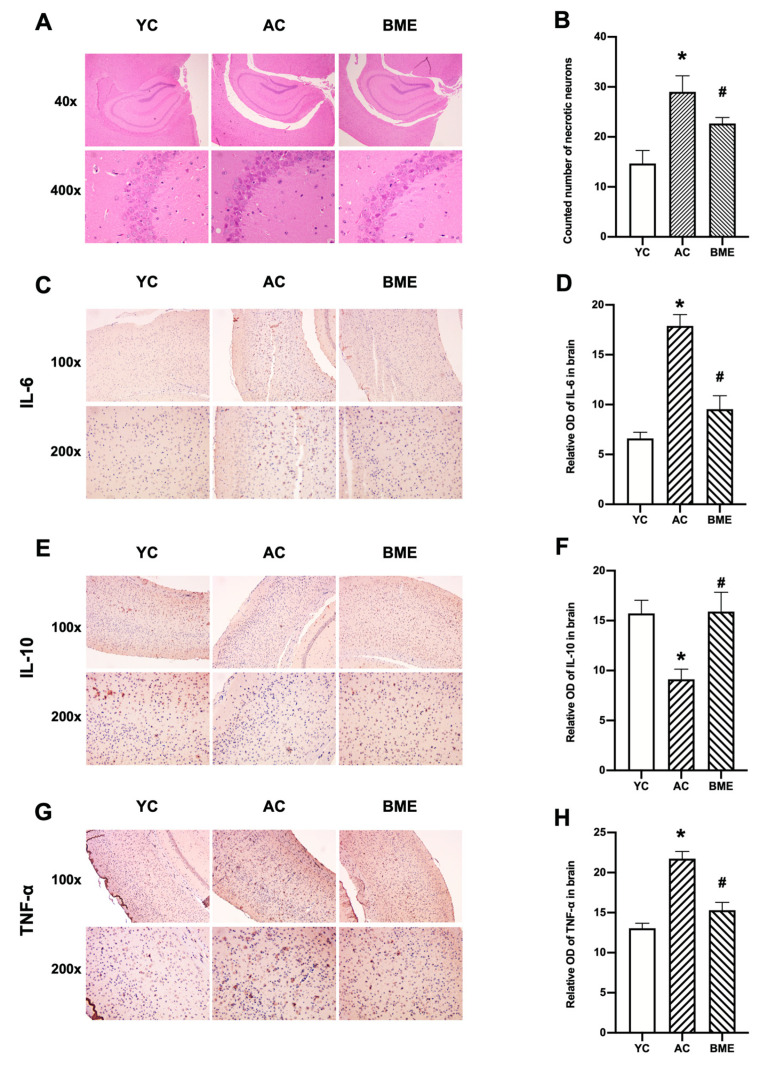
Histological morphology, immunohistochemical staining, and optical density (OD) values of cytokines in mouse brains. (**A**) HE staining of mouse brains. (**B**) The necrotic neurons per field in the CA3 region were counted. Representative micrographs of (**C**) IL-6, (**E**) IL-10, and (**G**) TNF-α immunostaining visualized with DAB and counterstained with hematoxylin. Integrated OD values of (**D**) IL-6, (**F**) IL-10, and (**H**) TNF-α were used to analyze the results of the immunohistochemistry. All results are expressed as means ± SEMs (*n* = 3). * *p* < 0.05 versus YC group, # *p* < 0.05 versus AC group. AC: aged control, BME: blueberry–mulberry extracts, YC: young control.

**Figure 3 foods-12-00860-f003:**
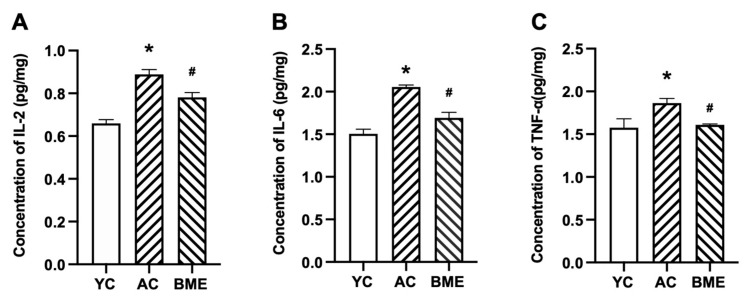
Effects of BME on the expression levels of the cytokines (**A**) IL-2, (**B**) IL-6, and (**C**) TNF-α in mouse brain tissues were detected using ELISA. All results are expressed as means ± SEMs (*n* = 6). * *p* < 0.05 versus YC group, # *p* < 0.05 versus AC group. AC: aged control, BME: blueberry–mulberry extract, YC: young control.

**Figure 4 foods-12-00860-f004:**
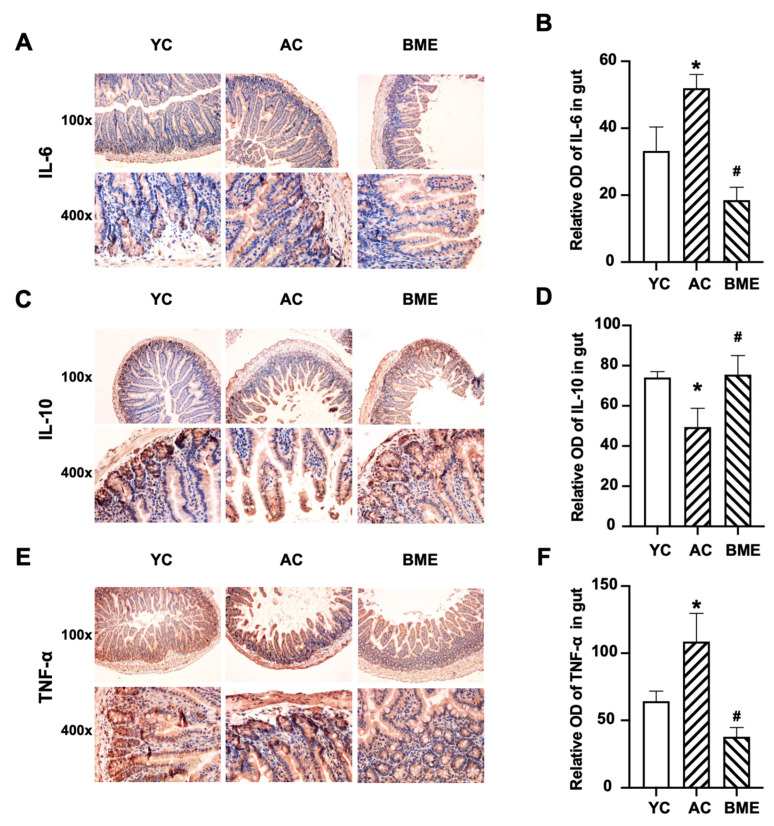
Immunohistochemical staining and optical density (OD) values of cytokines in mouse intestines. Representative micrographs of (**A**) IL-6, (**C**) IL-10, and (**E**) TNF-α immunostaining visualized with DAB and counterstained with hematoxylin. Integrated OD values of (**B**) IL-6, (**D**) IL-10, and (**F**) TNF-α were used to analyze the results of the immunohistochemistry. All results are expressed as means ± SEMs (*n* = 3). * *p* < 0.05 versus YC group, # *p* < 0.05 versus AC group. AC: aged control, BME: blueberry–mulberry extract, YC: young control.

**Figure 5 foods-12-00860-f005:**
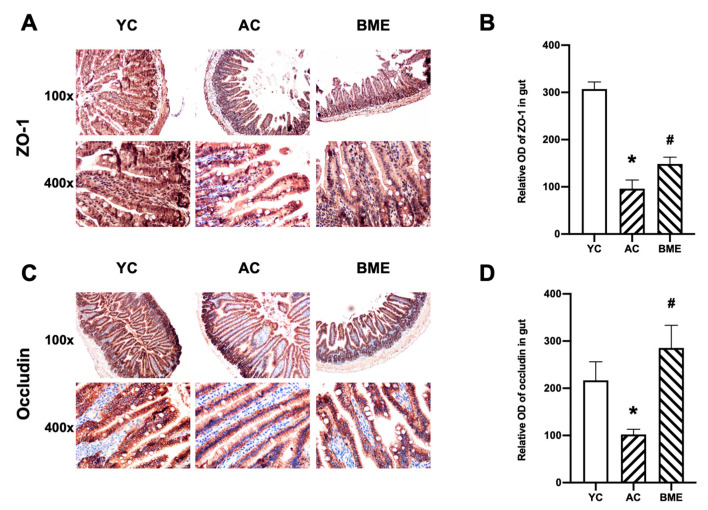
Immunohistochemical staining and optical density (OD) values of intestinal tight junction proteins in mouse intestines. Representative micrographs of (**A**) ZO-1 and (**C**) occludin immunostaining visualized with DAB and counterstained with hematoxylin. Integrated OD values of (**B**) ZO-1 and (**D**) occludin were used to analyze the results of the immunohistochemistry. All results are expressed as means ± SEMs (*n* = 3). * *p* < 0.05 versus YC group, # *p* < 0.05 versus AC group. AC: aged control, BME: blueberry–mulberry extract, YC: young control, ZO-1: zonula occludens-1.

**Figure 6 foods-12-00860-f006:**
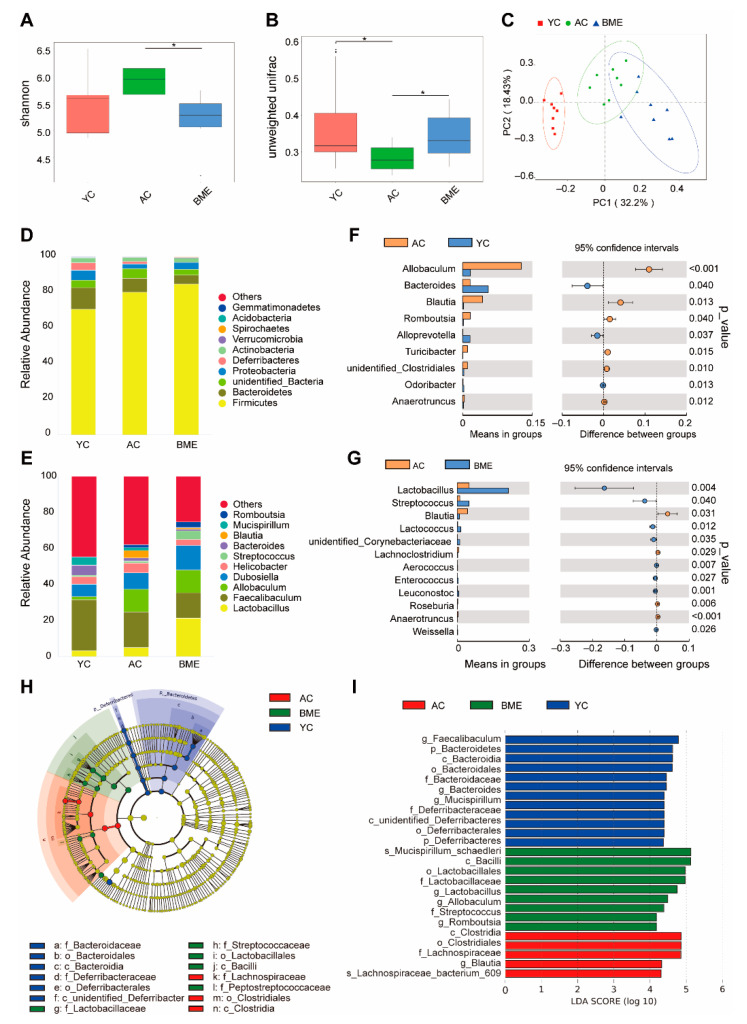
Effects of BME on the gut microbiota of mice were detected using a 16S rRNA sequencing-based analysis. (**A**) Alpha diversity of the microbial community in each group, analyzed by the Shannon index. (**B**) Intragroup beta diversity of the microbial community in each group, measured by unweighted UniFrac. (**C**) The principal components analysis based on Bray–Curtis metrics. (**D**) Relative abundances of predominant bacteria at the phylum level. (**E**) Relative abundances of top 10 bacteria at the genus level. (**F**) Differential microbiota analysis between the AC and YC groups. (**G**) Differential microbiota analysis between the AC and BME groups. (**H**) LEfSe analysis performed among the three groups. (**I**) Cardiogram showing differentially abundant taxonomic clades with LDA scores of 4.0 among groups with *p* values of 0.05. All results are expressed as means ± SEMs (*n* = 8). * *p* < 0.05 between groups. YC: young control, AC: aged control, BME: blueberry–mulberry extracts, LDA: linear discriminant analysis, LEfSe: linear discriminant analysis effect size, YC: young control.

**Figure 7 foods-12-00860-f007:**
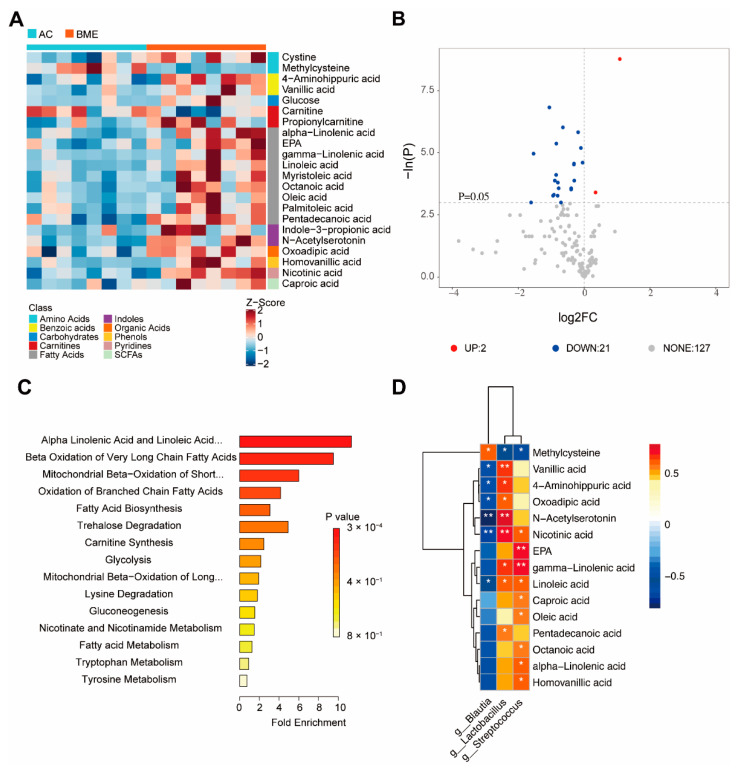
Effects of BME on metabolites of intestinal microbiota of mice were detected by targeted metabolomic profiling. (**A**) A heat map of intestinal metabolite levels. (**B**) A volcanic map of intestinal metabolite levels. (**C**) Differential metabolite enrichment analysis. (**D**) A correlation heat map between concentrations of differential metabolites and intestinal microbiota. All results are expressed as means ± SEMs (*n* = 8). * *p* < 0.05 versus AC group, ** *p* < 0.01 versus AC group. AC: aged control, BME: blueberry–mulberry extract.

**Table 1 foods-12-00860-t001:** Ingredient composition of the experimental diet.

Ingredient (g/kg)	Diet
Corn starch	465.692
Maltodextrin	155
Casein	140
Sucrose	100
Cellulose	50
Soybean oil	40
Mineral mix *	35
Vitamin mix *	10
Choline bitartrate	2.5
L-cystine	1.8
Tert butyl hydroquinone	0.008
Total (g)	1000
Energy (kcal/g)	3.81
Carbohydrate (% energy)	76.7
Protein (%)	13.6
Fat (%)	9.7

* The mineral and vitamin mixtures of the diets followed AIN-93M for rodents in the maintenance phase.

## Data Availability

The data are contained within the article.

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
