# Peer review of "Blueberry–Mulberry Extract Alleviates Cognitive Impairment, Regulates Gut Metabolites, and Inhibits Inflammation in Aged Mice"

_foods, 2023, doi:10.3390/foods12040860_

Round 1

Reviewer 1 Report

Extensive language editing is required, and it is quite difficult to understand what the authors meant.

Please include the catalog number of all the chemicals used in the study.

Figure 1 legend seems to be incorrect. For figs 1C, D, and E. Are authors sure about the p-value? By looking at the error bar it doesn’t seem so. Please have a look.

Based on what observation authors are concluding as apoptotic neurons in Fig 2B. In the text, the authors are suggesting it to be necrotic neurons. How many foci were considered for the counting, was it a total tissue per slide? Please explain.

The authors should either try to explain the text as per the figure arrangement or vice versa.  

How the OD was measured and the sample processed? There is no information.

In explaining the result, please mention the technique used to obtain the result.

Author Response

Dear reviewer:

Thank you for your comments and we have revised the manuscript according to your suggestions. The details were listed as follows:

Q1 Extensive language editing is required, and it is quite difficult to understand what the authors meant.

Response: Thanks for the reviewer's suggestion, we have revised the manuscript carefully. For example, the title was changed to “Blueberry-mulberry Extract Alleviates Cognitive Impairment, Regulates Gut Metabolites, and Inhibits Inflammation in Aged Mice”. For another instance, the sentence in lines 119-121 was revised to “The number of times for the mouse passed across the platform and the time spent in the target quadrant were recorded.” In addition, we have made many changes, which were highlighted in red in the manuscript.

Q2 Please include the catalog number of all the chemicals used in the study.

Response: Thanks for your suggestion. We have added catalog number of primary antibodies (line151-153) and ELISA kits (line 132-135) in manuscript.

Q3 Figure 1 legend seems to be incorrect. For figs 1C, D, and E. Are authors sure about the p-value? By looking at the error bar it doesn’t seem so. Please have a look.

Response: According to the suggestion of reviewer, we have revised the manuscript and all the data were presented as means ± SEM.

Q4 Based on what observation authors are concluding as apoptotic neurons in Fig 2B. In the text, the authors are suggesting it to be necrotic neurons. How many foci were considered for the counting, was it a total tissue per slide? Please explain.

Response: The text in Fig.2 was revised to “necrotic neurons”. To assess necrotic neurons in the CA3 region, shrunken neurons with pyknotic nuclei were considered as necrotic neurons. The Image Pro Plus software (version 6.0) was used to count the number of necrotic neurons per field under a light microscope. Data were expressed as the number of necrotic neurons/field, as previously reported [1]. Method for counting the number of necrotic neurons has been added to the manuscript (line140-145).

Q5 The authors should either try to explain the text as per the figure arrangement or vice versa.  

Response: We have changed the order of these figures (line 225-228) and revised legends for all the figures. In order to make the result clearer, we describe the result in the manuscript mainly according to the narrative logic.

Q6 How the OD was measured and the sample processed? There is no information.

Response: The digital images were analyzed using Image Pro Plus software (version 6.0), as previously described [2, 3]. The gray unit was switched to the optical density (OD) unit in the software, and the integrated OD analysis was performed with immunohistochemical stained regions. The methods were added to the manuscript (line 158-162).

Q7 In explaining the result, please mention the technique used to obtain the result.

Response: We have revised the results as follows: “We performed immunohistochemistry on the cerebral cortex and ELISA on whole brain lysates to investigate the effects of aging and BME treatment on inflammation.” (line 249-250) and now all the results begin with the technique we used. In addition, we also revised the legends for all the figures.

References

[1] Xu H, Gao H, Zheng W, Xin N, Chi Z, Bai S, et al. Lactational Zinc Deficiency-Induced Hippocampal Neuronal Apoptosis by a BDNF-independent TrkB Signaling Pathway. Hippocampus. 2011; 21(5):495-501. doi: 10.1002/hipo.20767.

[2] Sun W, Zhu M, Li J, Zhang X, Liu Y, Wu X, et al. Effects of Xie-Zhuo-Chu-Bi-Fang on miR-34a and URAT1 and Their Relation-ship in Hyperuricemic Mice. J Ethnopharmacol. 2015; 161:163-9. doi: 10.1016/j.jep.2014.12.001.

[3] Tung C, Ho C, Hsu Y, Huang S, Shih Y, Lin C. MicroRNA-29a Attenuates Diabetic Glomerular Injury through Modulating Cannabinoid Receptor 1 Signaling. Molecules. 2019; 24(2):264. doi: 10.3390/molecules24020264.

Reviewer 2 Report

The title could be changed as follows: Blueberry-mulberry Extract Alleviates Cognitive Impairment, Regulates Gut Metabolites, and Inhibits Inflammation in Aged Mice

The open-field test lacks positive control.

All the results could be presented as mean ± SEM.

In lines 105-106, there is a grammatical mistake.

Discussion section. 

a) The authors mentioned that BME restored intestinal epithelial barrier function. How was this effect evaluated? Was permeability or other functions assessed intestinal epithelial barrier?

b) The chemical composition of BME could be compared with other studies.

c) Does any of the compounds identified in BME exert reported anti-inflammatory actions?

Author Response

Dear reviewer:

Thank you for your comments and we have revised the manuscript according to your suggestions. The details were listed as follows:

Q1 The title could be changed as follows: Blueberry-mulberry Extract Alleviates Cognitive Impairment, Regulates Gut Metabolites, and Inhibits Inflammation in Aged Mice.

Response: The title was changed to “Blueberry-mulberry Extract Alleviates Cognitive Impairment, Regulates Gut Metabolites, and Inhibits Inflammation in Aged Mice”.

Q2 The open-field test lacks positive control.

 Response: In pharmaceutical research, it is very necessary to set positive control group in behavioral experiments (including MWM and open field tests). However, due to the long intervention period of nutrition research and most intervention material come from food, positive control group in behavioral experiments are rarely designed. Recently, many studies reported that the consumption of anthocyanins delayed brain aging process [1]. Furthermore, our experiment determined whether berry extract could reduce age-related brain declines in mice after nutritional intervention. We aimed to serve as a basis to provide a new strategy for nutritional intervention against aging-related cognitive decline in the elderly. Therefore, we didn’t set up a positive control group in this experiment.

Q3 All the results could be presented as mean ± SEM.

 Response: All the results were revised and were presented as mean ± SEM.

Q4 In lines 105-106, there is a grammatical mistake.

Response: The sentence in lines 119-121 was revised to “The number of times for the mouse passed across the platform and the time spent in the target quadrant were recorded.”

Q5 The authors mentioned that BME restored intestinal epithelial barrier function. How was this effect evaluated? Was permeability or other functions assessed intestinal epithelial barrier?

Response: ZO-1 and Occludin are the tight junction proteins of intestinal epithelial, and their contents reflect intestinal epithelial barrier function [2]. To evaluate the intestinal permeability, we detected the levels of the tight junction proteins using immunohistochemistry. The results showed that the levels of ZO-1 and occludin were inhibited by aging (Fig. 5, p < 0.05), suggesting an abnormality in intestinal permeability. BME treatment elevated the expression levels of ZO-1 and occludin in aged mice (p < 0.05), suggesting that BME reduced intestinal inflammation and protected intestinal permeability in these mice. These descriptions could be found in the results section (line 283-289) of the manuscript.

Q6 The chemical composition of BME could be compared with other studies.

 Response: Anthocyanins are water-soluble flavonoid pigments widely found in plants, and their types and concentrations vary with the fruit species and areas of origin, giving petals and fruits a variety of colors. Yang et al. reported that blueberry extract contains various anthocyanins, including cyanidin-3-O-glucoside (Cy3G), malvidin-3-O-glucose, pelargonidin-3-O-galactoside, and delphinidin-3-glucoside [3]. Mo et al. found the main components of mulberry extract were Cy3G and cyanidin-3-O-rutinoside [4]. In our previous study, the content of Cy3G in blueberry extract was about 25% and that in mulberry extract was 6.8% [5,6]. The biological activities of anthocyanins are closely related to their structures [7]. The mixing of two extracts with variety of anthocyanins may be beneficial to their biological function. These descriptions were added in the discussion (line 478-487) of the manuscript.

Q7 Does any of the compounds identified in BME exert reported anti-inflammatory actions?

 Response: Numerous studies have described that anthocyanins, the main content of BME, may play a crucial role in the prevention and treatment of different pathological conditions, which have encouraged their consumption around the world [8]. Anthocyanins exhibit a significant neuroprotective role, mainly due to their well-recognized antioxidant and anti-inflammatory properties. However, which anthocyanin monomer is beneficial for cognitive improvement needs further research. These descriptions were added in the discussion (line 388-398) of the manuscript.

References

[1] Jennings A, Steves CJ, Macgregor A, Spector T, Cassidy A. Increased Habitual Flavonoid Intake Predicts Attenuation of Cognitive Ageing in Twins. BMC Med. 2021; 19(1):185. doi: 10.1186/s12916-021-02057-7.

[2] Kuo W, Odenwald MA, Turner JR, Zuo L. Tight Junction Proteins Occludin and ZO-1 as Regulators of Epithelial Proliferation and Survival. Ann N Y Acad Sci. 2022; 1514(1):21-33. doi: 10.1111/nyas.14798.

[3] Yang S, Wang C, Li X, Wu C, Liu C, Xue Z, et al. Investigation on the Biological Activity of Anthocyanins and Polyphenols in Blueberry. J Food Sci. 2021; 86(2):614-627. doi: 10.1111/1750-3841.15598.

[4] Mo J, Ni J, Zhang M, Xu Y, Li Y, Karim N, et al. Mulberry Anthocyanins Ameliorate DSS-Induced Ulcerative Colitis by Improving Intestinal Barrier Function and Modulating Gut Microbiota. Antioxidants (Basel). 2022; 11(9):1674. doi: 10.3390/antiox11091674.

[5] Song N, Yang H, Pang W, Qie Z, Lu H, Tan L, et al. Mulberry Extracts Alleviate Aβ 25-35-induced Injury and Change the Gene Expression Profile in PC12 Cells. Evid Based Complement Alternat Med. 2014; 2014:150617. doi: 10.1155/2014/150617.

[6] Li H, Zheng T, Lian F, Xu T, Yin W, Jiang Y. Anthocyanin-rich Blueberry Extracts and Anthocyanin Metabolite Protocatechuic Acid Promote Autophagy-lysosomal Pathway and Alleviate Neurons Damage in in vivo and in vitro Models of Alzheimer's Disease. Nutrition. 2022; 93:111473. doi: 10.1016/j.nut.2021.111473.

[7] Rahman MM, Ichiyanagi T, Komiyama T, Hatano Y, Konishi T. Superoxide Radical- and Peroxynitrite-scavenging Activity of Anthocyanins; Structure-activity Relationship and Their Synergism. Free Radic Res. 2006; 40(9):993-1002. doi: 10.1080/10715760600815322.

[8] Henriques JF, Serra D, Dinis TCP, Almeida LM. The Anti-Neuroinflammatory Role of Anthocyanins and Their Metabolites for the Prevention and Treatment of Brain Disorders. Int J Mol Sci. 2020; 21(22):8653. doi: 10.3390/ijms21228653.

Reviewer 3 Report

As the world population ages, many age-related diseases can be found in the elderly. Cognitive impairment, a major disorder in this age-group, is associated with low grade chronic neuroinflammation, imbalanced redox processes, damaged neurogenesis, mitochondrial dysfunction, synaptic impairment leading to the development of neurodegenerative diseases, especially dementia that gravely decreases the quality of life.

Recent studies increasingly emphasize the role gut microbiota could play in modulating the brain function and behavior through the gut-brain axis and the fact that nutritional interventions ameliorate the composition of gut microbiota in the whole population but mostly in the elderly. Berries, plant-source foods very rich in polyphenols, demonstrated to alter gut microbiome and contribute to the relief of inflammation and cognitive impairment. 

The present study exposed that blueberry-mulberry extracts could alleviate cognitive decline in aged mice, analyzed the gut microbiota-derived metabolites, behavioral phenotypes, and histopathology of the brain in treated mice.

The topic is important and the manuscript provides a comprehensive analysis of the subject. However, the following suggestions should be addressed:

- English needs to be revised in several paragraphs

- there is no mention of the extract in Methods! How were the blueberry-mulberry extracts made? What plant matrices were used? When were these collected?

Lines 368-370: a very recent study mentions that berries, ellagitannin-rich fruits, through gut microbiota-derived metabolites, could modulate antioxidant and anti-inflammatory signaling pathways, scavenge free radicals and ROScontribute to the intestinal wall integrity, act on the gut–brain axis, and promote beneficial health effects (Banc et al. doi: 10.3390/foods12020270).

- the references should be updated; all the studies cited in this research were published on 2020 or before. A quick search on Pubmed revealed around 30 articles related to blueberry/mulberry-inflammation and 5 articles related to blueberry/mulberry-cognition in mice published in the last two years, many in MDPI journals, and at least a few of them should have been used as references and discussed. Please see:

doi: 10.3390/foods11121818

doi: 10.3390/nu14132734

doi: 10.3390/ijms22158120

doi: 10.1016/j.nut.2021.111473

doi: 10.3390/molecules26040920

doi: 10.3390/ijms23169259

doi: 10.1007/s11064-022-03813-8

doi: 10.3390/molecules27010108

doi: 10.1111/gtc.12889

Author Response

Dear reviewer:

Thank you for your comments and we have revised the manuscript according to your suggestions. The details were listed as follows:

Q1 English needs to be revised in several paragraphs

Response: Thanks for the reviewer's suggestion, we have revised the manuscript carefully. For example, the title was changed to “Blueberry-mulberry Extract Alleviates Cognitive Impairment, Regulates Gut Metabolites, and Inhibits Inflammation in Aged Mice”. For another instance, the sentence in line 119-121 was revised to “The number of times for the mouse passed across the platform and the time spent in the target quadrant were recorded.” In addition, we have made many changes, which were highlighted in red in the manuscript.

Q2 there is no mention of the extract in Methods! How were the blueberry-mulberry extracts made? What plant matrices were used? When were these collected?

Response: The methods and matrices for blueberry-mulberry extracts were added in “2.1 Materials” (line 81-91). “Blueberry fruits (Vaccinium uliginosum L.) were purchased from Daxinganling Lingongberry Organic Foodstuffs Co. Ltd., Harbin, Heilongjiang Province, China. Fresh blueberry fruits were crushed and extracted four times with fresh ethanol (75%) at room temperature for 96 h. The four ethanolic phases were recovered, pooled, and dried at 40°C in a rotary evaporator and lyophilized [1]. Mulberry fruits (Morus nigra L.) were purchased from Xinjiang Bencaotang Traditional Chinese Herbal Decoction Pieces Co. Ltd., Turfan, Xinjiang Uygur Autonomous Region, China. Blended fresh mulberry fruits were extracted five times with 60% alcohol and dehydrated in a rotary evaporator and lyophilized [2]. BME stock solution was prepared by dissolving blueberry extract and mulberry extract (1:1) in water for gavage.”

Q3 Lines 368-370: a very recent study mentions that berries, ellagitannin-rich fruits, through gut microbiota-derived metabolites, could modulate antioxidant and anti-inflammatory signaling pathways, scavenge free radicals and ROS, contribute to the intestinal wall integrity, act on the gut–brain axis, and promote beneficial health effects (Banc et al. doi: 10.3390/foods12020270).

Response: Thanks for the reviewer's suggestion, and the reference was added in line 574-575.

Q4 the references should be updated; all the studies cited in this research were published on 2020 or before. A quick search on Pubmed revealed around 30 articles related to blueberry/mulberry-inflammation and 5 articles related to blueberry/mulberry-cognition in mice published in the last two years, many in MDPI journals, and at least a few of them should have been used as references and discussed. Please see:

doi: 10.3390/foods11121818

doi: 10.3390/nu14132734

doi: 10.3390/ijms22158120

doi: 10.1016/j.nut.2021.111473

doi: 10.3390/molecules26040920

doi: 10.3390/ijms23169259

doi: 10.1007/s11064-022-03813-8

doi: 10.3390/molecules27010108

doi: 10.1111/gtc.12889

Response: Thanks for the reviewer's suggestion, and the references were updated and some of them were added in line 626,629, and 632.

References

[1] Tan L, Yang H, Pang W, Li H, Liu W, Sun S, et al. Investigation on the Role of BDNF in the Benefits of Blueberry Extracts for the Improvement of Learning and Memory in Alzheimer’s Disease Mouse Model. J Alzheimers Dis. 2017; 56:629-640. doi: 10.3233/JAD-151108.

[2] Song N, Yang H, Pang W, Qie Z, Lu H, Tan L, et al. Mulberry Extracts Alleviate Aβ 25-35-induced Injury and Change the Gene Expression Profile in PC12 Cells. Evid Based Complement Alternat Med. 2014; 2014:150617. doi: 10.1155/2014/150617.

Round 2

Reviewer 1 Report

The authors have made a substantial improvement in the revised manuscript and it is certainly will be interesting to readers. However, the author should clarify about the data availability statement as the raw data associated with sequencing and metabolomics data should be made available either through supplementary files or shared upon reasonable request. Please check the journal policy.

Reviewer 2 Report

The manuscript can be accepted for publication.

Reviewer 3 Report

The authors addressed all the suggestions and questions; the manuscript is much improved.

I have no further recommendations.